# Psychological, social and cognitive resources and the mental wellbeing of the poor

Deborah A. Cobb-Clark[1,2,3], Nathan Kettlewell [2,3,4]*

1 School of Economics, The University of Sydney, Sydney, NSW, Australia, 2 ARC Centre of Excellence for Children and Families over the Life Course, School of Economics, The University of Sydney, Sydney, NSW, Australia, 3 Institute of Labor Economics (IZA), University of Bonn, Bonn, Germany, 4 Economics Discipline Group, University of Technology Sydney, Sydney, NSW, Australia

* Nathan.Kettlewell@uts.edu.au

## Abstract

Our study takes advantage of unique data to quantify deficits in the psychosocial and cognitive resources of an extremely vulnerable subpopulation–those experiencing housing vulnerability–in an advanced, high-income country (Australia). Groups such as these are often impossible to study using nationally representative data sources because they make up a small share of the overall population. We show that those experiencing housing vulnerability sleep less well, have more limited cognitive functioning, and less social capital than do those in the general population. They are also less emotionally stable, less conscientious, more external, and more risk tolerant. Collectively, these deficits in psychosocial and cognitive resources account for between 24–42% of their reduced life satisfaction and their increased mental distress and loneliness. These traits also account for a large proportion of the gap in mental wellbeing across different levels of housing vulnerability.

## 1. Introduction

The way that people perceive the world, process information, and make decisions is shaped by not only by their beliefs and values, but also by their psychological, social, and cognitive resources. Maintaining healthy and stable psychological and physical functioning in the face of traumatic events, for example, rests not only on people's economic resources, but also on their personalities including their control perceptions [1]. Those with internal control perceptions experience less emotional distress following illness or injury, victimization, and bereavement [2,3]; they also return to work sooner following bad health shocks [4]. Many positive life outcomes including academic achievement and labor market success (e.g. [5,6]), good health (e.g. [7]), and financial wellbeing (e.g. [8]) can be directly linked to the personal psychosocial and cognitive resources that people have available to draw upon.

Poverty taxes these resources. Financial uncertainty is both emotionally and cognitively demanding. People's attention is directed solely towards meeting their immediate needs (food, shelter), limiting their ability to focus on longer-run issues [9–11]. Meanwhile, stress is compounded by the fact that the poor operate in more menacing and less co-operative environments [12]. Many of the risks they face cannot be diversified or insured [13]. "Extreme poverty deprives people of almost all means of managing risk by themselves" [14].

meet the criteria for access to confidential data. Our study uses third party data from the Journeys Home Survey and the Household, Income and Labour Dynamics in Australia (HILDA) Survey. Data from the Journeys Home Survey is available for purchase under individual licensing arrangements with the Australian Government's Department of Social Services. For instructions on how to access the data, researchers should email JH@dss.gov.au. Further information is available at https://melbourneinstitute.unimelb.edu.au/journeys-home/for-researchers. Our study can be replicated using any of the data releases, including the International Release (we used the Limited Release). HILDA can be applied for at no cost to researchers through the National Centre for Longitudinal Data Dataverse. Instructions are available at https://dataverse.ada.edu.au/dataverse/hilda. We used the General Release 16; later releases should provide nearly identical results. We did not have special privileges to obtain the data and obtained our data through the process described above.

**Funding:** This research received financial support from the Australian Research Council through a Discovery Project Grant (DP140102614) and the Centre of Excellence for Children and Families over the Life Course (CE140100027).

**Competing interests:** The authors have declared that no competing interests exist.

It is not surprising then that poverty pushes vulnerable people towards less effective coping mechanisms. They are more likely to use passive, emotional, and evasive strategies for managing stress (see [15] for a review) and turn to costly forms of borrowing (e.g. pay-day loans) [16] to manage financial shortfalls. Disadvantage has also been linked to more external control tendencies [15], to more impatience, diminished self-control, and increased risk aversion [17,18] and to undesirable personality changes in children [19]. Poverty may intensify risk aversion resulting in a vicious cycle of disadvantage as the poor underinvest in their own human capital and financial security (see [17,20]). Alternatively, the disadvantaged may engage in riskier behavior because of concerns about relative economic position [21] or because they are unlikely to achieve their goals through low-risk means [22].

The goals of this study are twofold. First, we compare the psychological, social, and cognitive resources of a highly disadvantaged group to those of the general population in a high-income country (Australia). We also provide comparative evidence *within* the disadvantaged group by examining resources by degree of disadvantage. Second, we analyse the extent to which disparity in people's psychological, social, and cognitive resources contributes to diminished mental wellbeing.

Our research is strengthened by the unique data we analyse. Our sample of disadvantaged individuals is drawn from the universe of administrative social assistance records, while detailed survey data allow us to analyse three measures of mental wellbeing (mental health, life satisfaction, loneliness) accounting for a comprehensive set of psychological (emotional stability, conscientiousness, control perceptions), social (support network), and cognitive (working memory, risk affinity, future orientation, sleep quality) resources. The resources we consider–while quite distinctive–are all impacted by disadvantage and foundational to the way people make choices.

Our research is also novel in examining poverty through the lens of housing vulnerability. International law has recognized adequate housing as a basic human right for nearly three generations [23]; yet housing vulnerability remains a pressing issue in many wealthy countries. Homelessness increases the risk of psychological trauma [24] and is both a cause and a consequence of poverty.

We find large deficits for the poor across the range of personal resources we consider. To demonstrate the relative importance of these deficits, we conduct a decomposition analysis and show that they account for between 24–42% of the reduced cognitive wellbeing, increased mental distress, and increased loneliness of people who are currently, or were recently, homeless. Across our various wellbeing measures, this is always more than or similar to the combined effect of demographic and family background characteristics.

## 2. Data

### 2.1 Journeys Home and Household, Income and Labour Dynamics in Australia data

This study draws on data from the Journeys Home Project in which a representative sample of housing-vulnerable Australians was selected using the Australian government's protocols for flagging all social assistance recipients thought to be either 'homeless' or 'at risk of homelessness'. Virtually all Australians in vulnerable housing situations, e.g. couch surfing, public housing, shelters, boarding houses, living on the streets, etc. receive some form of social assistance; thus, the Journeys Home sampling frame results in a much broader representation of the homeless population than do previous studies. Sampled individuals were subsequently interviewed about their circumstances, including their housing histories, in six bi-annual waves (2011–2014). More detail on the design of the Journeys Home Project and its fieldwork

outcomes can be found in [25]. Our analysis draws on the 80 percent (n = 1,138 in wave 5) of those who report at least one episode of homelessness during the study period. Homelessness means involuntarily living without conventional accommodation; or living in a caravan, hotel, boarding house, crisis accommodation; or living with friends, family or other relatives, either temporarily, through welfare service provision or without a separate bedroom (see S1 Appendix in S1 Online Appendix for further details).

Journeys Home respondents can be reasonably thought of as a draw from the 'small segment of the population with large economic burden' [26]. To understand how housing vulnerability interacts with people's psychological, social, and cognitive resources, we compare the characteristics of Journeys Home respondents to those of a representative sample drawn from the Household, Income and Labour Dynamics in Australia (HILDA) Survey. HILDA is a representative longitudinal survey that was launched in 2001 and has followed a panel of Australian households annually since (see [27]).

## 2.2 Estimation sample and variable construction

Our analysis pools the Journeys Home and HILDA datasets to make comparisons across these populations. Our main variables are measures of psychosocial resources and mental wellbeing that were collected in both surveys. We face two challenges with comparing these variables across the samples. First, the time periods in which some variables were collected do not always line up. The biggest difference is two years (for our working memory measure). For our decomposition analysis, where we need controls for all the resource measures, we take the measures from whatever year they are available and treat them as if they are from the focus year (2014). This approach is supported by literature on the temporal stability of these resources in the Australian working-age population [28,29].

The second challenge is that there are differences in the way that some variables are measured in the two surveys. For example, in HILDA the personality questions use a 7-point scale, whereas Journeys Home uses a 5-point scale. In Table A1 in S1 Appendix in S1 Online Appendix we outline how each resource is measured in the respective surveys, and how we reconstruct them to achieve comparability. Where applicable, our scales were assessed for reliability by calculating ordinal alpha [30]; the values are all in the acceptable range (0.73–0.94).

## 2.3 Sample characteristics

Those vulnerable to homelessness are more likely than the average Australian to be male, Indigenous, native-born, and to have not completed high school (Table 1). They are more than a decade younger on average (age 32 vs. 46); almost three times as likely to be single; and nearly a third more likely to have a long-term health condition. Family background also matters for housing vulnerability. Nearly half of Journeys Home respondents have parents who were never together or who had separated by the time they were aged 16 –a rate that is almost three times that in the Australian population generally. Further details on our sample and variable definitions are in S1 Appendix in S1 Online Appendix. All figures and estimation results in this paper use population weights (supplied with the respective surveys) unless explicitly stated otherwise. Weights for the Journeys Home sample adjust differentially for three levels of housing vulnerability in the Australian general population (homeless, at risk and vulnerable). See [31] for further details.

## 3. Disparity in psychological, social and cognitive resources

There are many reasons to believe that housing vulnerability may deplete the psychosocial and cognitive resources available to people. Poor sleep, for example, is often a consequence of

**Table 1. Descriptive statistics comparison: Journeys Home and HILDA 2013.**

|  | Journeys Home[a] | HILDA[b] | Difference | p-value |
|---|---|---|---|---|
| Age | 31.93 | 46.29 | -14.36 | 0.00 |
| Male | 0.62 | 0.48 | 0.14 | 0.00 |
| ATSI | 0.18 | 0.02 | 0.16 | 0.00 |
| Student | 0.10 | 0.07 | 0.03 | 0.01 |
| University | 0.04 | 0.27 | -0.23 | 0.00 |
| Diploma | 0.07 | 0.09 | -0.03 | 0.05 |
| Certificate 3 or 4 | 0.29 | 0.22 | 0.07 | 0.00 |
| Year 12 | 0.10 | 0.17 | -0.07 | 0.00 |
| Married | 0.03 | 0.53 | -0.49 | 0.00 |
| Defacto | 0.19 | 0.12 | 0.08 | 0.00 |
| Separated | 0.08 | 0.03 | 0.06 | 0.00 |
| Divorced | 0.09 | 0.06 | 0.03 | 0.01 |
| Single | 0.59 | 0.22 | 0.37 | 0.00 |
| Aus. Born | 0.86 | 0.69 | 0.17 | 0.00 |
| Born main English | 0.06 | 0.11 | -0.05 | 0.00 |
| Long term health cond. | 0.38 | 0.30 | 0.08 | 0.00 |
| Parents separated age 16 | 0.44 | 0.16 | 0.28 | 0.00 |
| Parents never together | 0.05 | 0.01 | 0.04 | 0.00 |
| Mother university degree | 0.08 | 0.10 | -0.03 | 0.02 |
| Major urban | 0.77 | 0.71 | 0.06 | 0.00 |
| Other urban | 0.16 | 0.18 | -0.02 | 0.20 |
| Rural balance | 0.04 | 0.09 | -0.04 | 0.00 |

Notes

[a] n = 1,138.

[b] n = 16,432.

Columns show the population weighted means for each dataset for fully responding survey participants, their differences and the asymptotic p-values for whether the differences are significantly different from zero. ATSI indicates Aboriginal–Torres Strait Islander. See Table A2 in S1 Appendix in S1 Online Appendix for variable definitions.

inadequate housing [32]; and there is evidence that how well people sleep matters more for their mental wellbeing than how much they sleep [33]. [34] provide evidence that sleep is a resource supporting stress management and self-regulation, while [35] discuss sleep as a resource supporting productivity at work. Cognitive performance–and in particular, working memory–has also been found to be impaired in young people experiencing homelessness, foster care, or poverty [36]. Much of this research draws on small, highly selected samples with limited counter-factual evidence. One of the main contributions of our study is to provide population level comparisons.

## 3.1 Unconditional and conditional disparities in resources

We compare the distributions of the key psychosocial and cognitive resources of those who are housing vulnerable to the general population in Fig 1. (Detailed definitions for each of these measures are provided in Table A1 in S1 Appendix in S1 Online Appendix). We find that Australians experiencing housing vulnerability have significantly fewer psychological, social, and cognitive resources to draw on than do Australians generally. They sleep less well, have more limited cognitive functioning, less social capital and are less emotionally stable, less conscientious, and more external. Consistent with the risk sensitivity hypothesis [22], they are also

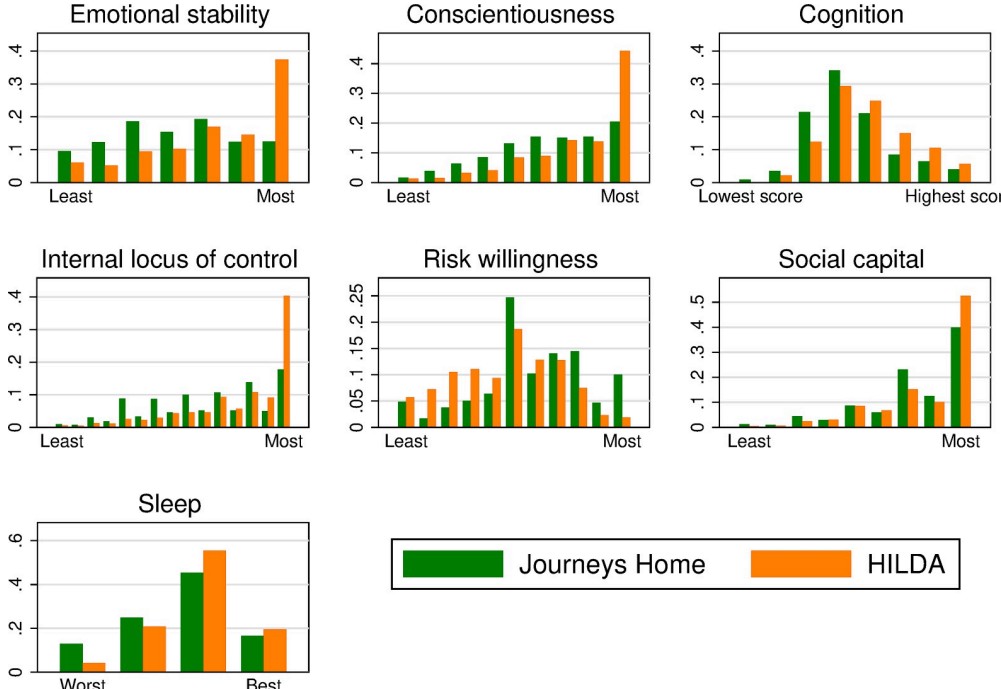

**Fig 1. Unconditional differences in psychological, social and cognitive resources.** Notes: Each panel shows frequencies of actual values. (a) The sum of agreement to the following personal characterizations measured on a three-point scale: Moody (reversed), touchy (reversed), temperamental (reversed). (b) The sum of agreement to the following personal characterizations measured on a three-point scale: Orderly; sloppy (reversed); disorganized (reversed); efficient. (c) Score (0–8) on a task to recall increasing numbers of digits in reverse order (test of working memory). (d) Sum of agreement to seven items regarding internality on a three-point scale. (e) Stated willingness to take risks in general (11-point scale). (f) Sum of agreement to four items regarding social support on a three-point scale. (g) Self-rated sleep quality (four-point scale).

more tolerant to risk. Table B1 in S2 Appendix in S1 Online Appendix reports the sample size used for each plot in Fig 1 and the mean difference (in standardized units) and its standard error.

Next, we investigate whether the psychological, social, and cognitive resources gaps we have identified in Fig 1 vary with the intensity of the housing vulnerability that Journeys Home respondents are experiencing. To this end, we follow [31] in defining four types of housing situations: i) homelessness; ii) insecure non-traditional housing; iii) secure non-traditional housing; and iv) housed (see Fig 2).

There is no single, commonly accepted definition of homelessness. Instead, notions of homelessness and housing insecurity more generally are usually derived from political considerations, statistical agency definitions, rules used to allocate public resources, or the data constraints faced by researchers (see [37]). Our categorization of housing situations corresponds to the varying definitions of homelessness developed by the Melbourne Institute [31]. Specifically, the Melbourne Institute categories follow the cultural definition of homelessness that assesses whether people's accommodation meets the minimum community standard that people can expect to achieve in contemporary Australian society [38].

Using the categories in Figs 2 and 3 depicts the resource gap (relative to the general HILDA population) across the entire housing vulnerability profile.

We find that across all housing situations, Journeys Home respondents generally have significantly fewer psychological, social, and cognitive resources than does the Australian

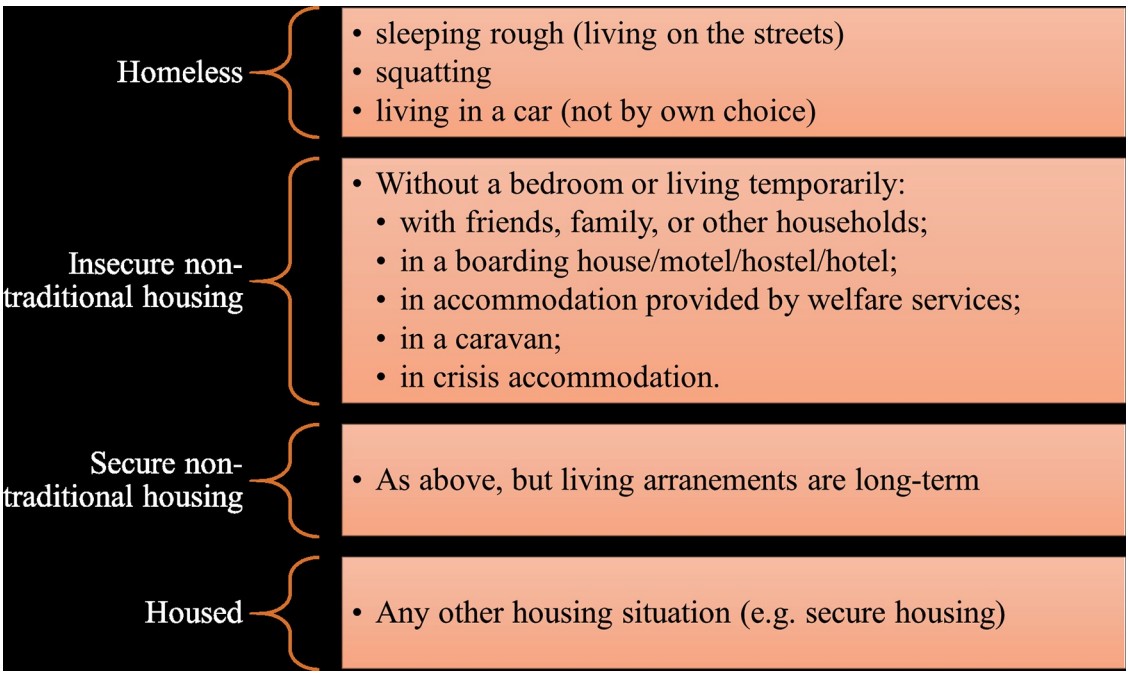

**Fig 2. Housing definitions.**

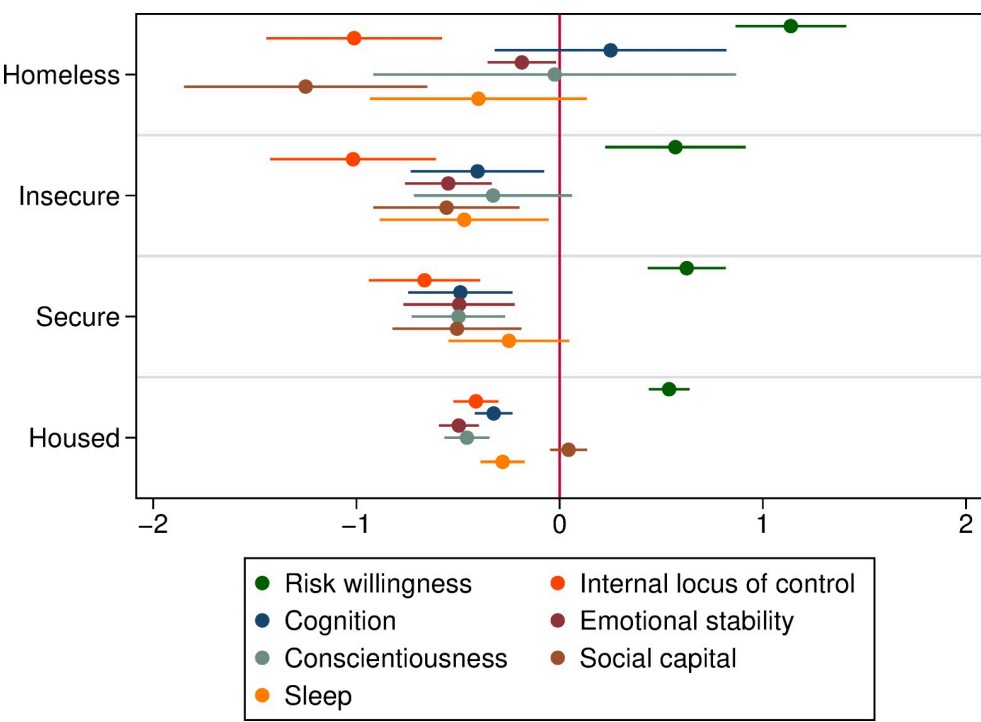

**Fig 3. Unconditional differences in psychological, social and cognitive resources by housing vulnerability.** (Journeys Home vs. HILDA Populations). Notes: Each point shows the mean difference (and 95% confidence interval) between the relevant housing vulnerability category in the Journeys Home sample compared to the HILDA sample. Depending on resource variable, there are 25–41 people in the Homeless group, 69–81 in Insecure, 118–141 in Secure and 778–875 in Housed.

population overall. Even those who are currently in secure housing arrangements have poorer sleep, and more limited cognitive functioning. They are also less emotionally stable, less conscientious, more external, and more tolerant to risk. The ubiquitous nature of the resource gap among the Journeys Home population–all of whom have experienced homelessness in the past–could indicate that housing insecurity has a permanent scaring effect on the resources that people are able to accumulate, although we cannot be sure about the direction of causality.

More intensive housing vulnerability is often linked to larger resource deficits within the Journeys Home population. Those sleeping rough, squatting, or living in their car have the largest deficits in perceptions of control, risk affinity, and social capital; all of which play a fundamental role in shaping the decisions that people make and the outcomes they achieve. People's locus of control and willingness to take risks drive their willingness to make health, education, and labor market investments in themselves and have consequences for their health outcomes, addictive behavior, financial decisions, and migration choices (see [39,40]).

We turn now to considering whether these differences in average resources across populations are explained by disparity in people's demographic characteristics (e.g. age, gender, marital status, etc.) and family background (e.g. family structure at age 16). Because the gaps in resources are generally consistent across homelessness status, and there are relatively few people in the 'homeless' and 'insecure' categories, we focus simply on the Journeys Home versus HILDA comparison. Specifically, we estimate a logistic regression model where the dependent variable is an indicator for being in the Journeys Home sample, and we include as independent variables all psychological, social, and cognitive resources (see Table A1 in S1 Appendix in S1 Online Appendix) and our control variables (see Table 1). We plot the resulting odds ratio estimates for the various resources we consider (and their 95% confidence intervals) in Fig 4.

Even after we take these controls into account, Journeys Home respondents have more limited personal resources (though the difference in sleep quality is not statistically significant) with the exception of social capital, which has the reversed sign. They also remain more risk tolerant. All odds ratios except that for sleep quality are significantly different from one.

Finally, one issue in comparing Journeys Home and HILDA data is that the populations necessarily overlap. We estimate, however, that less than 0.5% of HILDA respondents belong to the Journeys Home population (S3 Appendix in S1 Online Appendix). Reweighting the HILDA sample with predictions from a scaled binomial loss model [41,42] to form mutually exclusive groups does not affect our results (see Fig 4, weighted estimates).

## 3.2 The consequence of the resource gap for mental wellbeing

Constraints in the psychological, social, and cognitive resources that people can access are no doubt consequential for the choices they make, the outcomes they achieve, and their overall mental wellbeing. We consider this issue by investigating gaps in three dimensions of wellbeing: *affective wellbeing* captured by the Kessler-6 mental distress scale [43]; *cognitive wellbeing* (a global evaluation of one's life) captured by overall life satisfaction; and *loneliness* captured by self-reported experience. The prevalence of loneliness, and its concentration within certain groups, is increasingly viewed as an important health issue. Poor neighborhoods often have additional barriers to social engagement (e.g. inadequate housing, lack of transportation, fear of crime) that can contribute to heightened loneliness for their residents [44].

To explore the extent to which gaps in psychosocial resources 'explain' gaps in mental wellbeing, we conduct Blinder-Oaxaca decompositions [45,46]. Let $R = E(Y_{jh}) - E(Y_{hilda})$ be the

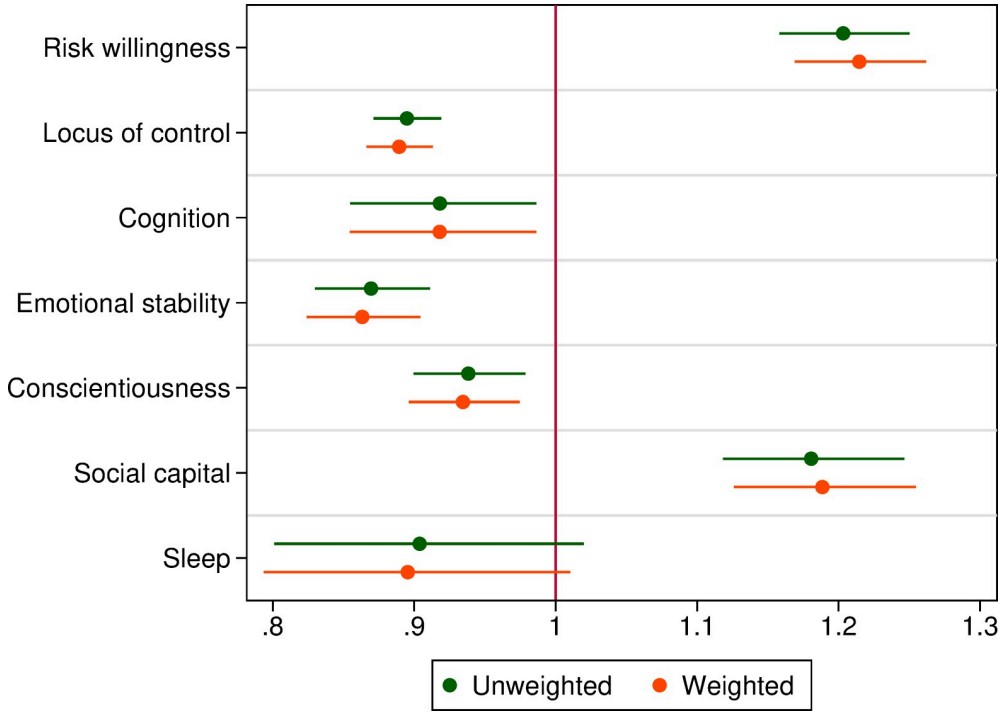

**Fig 4. Logit odds ratio estimates for being in the journeys home sample.** Notes: The dependent variable is an indicator for being in the Journeys Home sample. Estimates are from binary logit regressions and include all variables in Table A2 in S1 Online Appendix as additional controls. Error bars are 95% confidence intervals using robust asymptotic standard errors. The weighted estimates include weights for HILDA observations for the probability of not being in the severely disadvantaged population (see Section 2.3). Neither the weighted or unweighted estimates use population weights, since our estimates pool the Journeys Home and HILDA samples. $n = 11{,}637$ ($n_{hilda} = 10{,}798$, $n_{jh} = 839$).

unconditional mean gap in outcome $Y$ between the Journeys Home and HILDA populations. $R$ can be written as:

$$[E(X_{jh}) - E(X_{hilda})]'\beta_{hilda} + E(X_{jh})'[\beta_{jh} - \beta_{hilda}] \tag{1}$$

In Eq (1), $X_g$ is the vector of covariates (psychosocial resources and other controls) for group $g$ and $\beta_g$ are the coefficients from a linear regression on $Y$ for group $g$. $[E(X_{jh}) - E(X_{hilda})]'\beta_{hilda}$ is interpreted as the part of $R$ explained by differences in endowments (characteristics) evaluated at the return to those endowments estimated from a linear regression for the HILDA group ($\hat{\beta}_{hilda}$). This is consistent with a counterfactual thought experiment in which we are interested in how wellbeing would change if we could give the housing insecure the same resources as the general population, and the same returns on those resources (in Table D3 (Fig D2) in S4 Appendix in S1 Online Appendix we show results where $\hat{\beta}_{jh}$ is used instead, in which case psychosocial resources explain even more of the gaps in wellbeing). The second part of Eq (1) – $E(X_{jh})'[\beta_{jh} - \beta_{hilda}]$ – is the 'unexplained' gap, which can be interpreted as the gap due to differences in returns to endowments, although it is likely to also capture effects from unobserved factors.

Our decomposition results are summarized in Fig 5. The detailed decomposition estimates are also reported in S4 Appendix Table D1 in S1 Online Appendix, showing the individual contributions of each resource and other controls. People who are housing insecure have substantially poorer wellbeing than the general population. They experience more mental distress

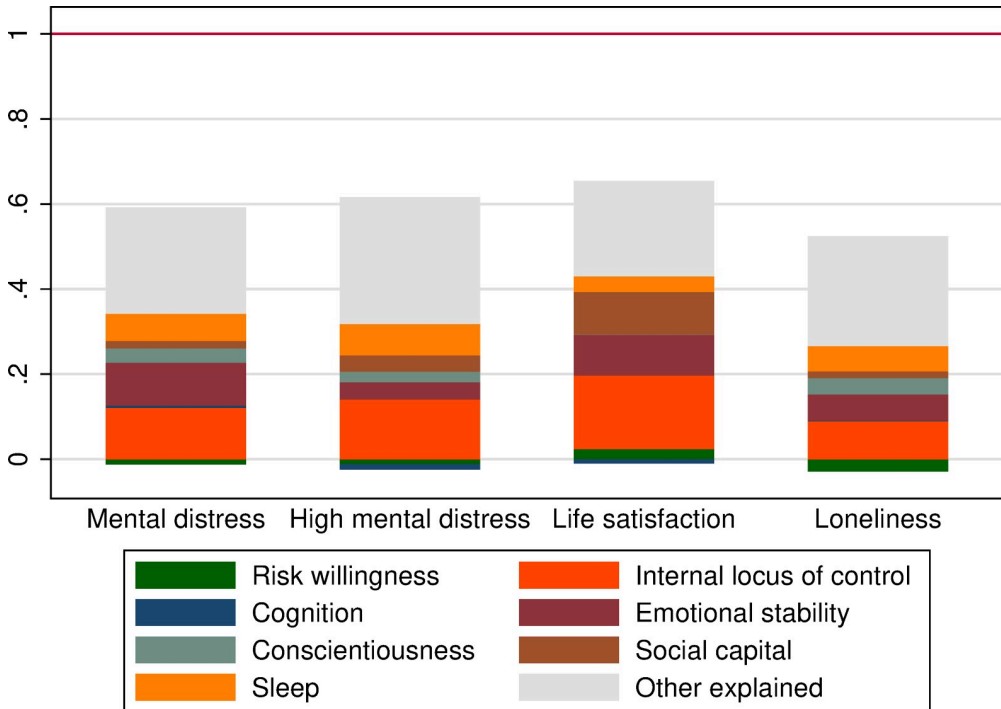

**Fig 5. Decomposition of the aggregate mental wellbeing gap.** (Journeys Home vs. HILDA Populations). Notes: Red lines show the population weighted unconditional gap between the Journeys Home and HILDA samples normalized to one. Stacked rectangles show the amount of the gap explained by differences in endowments of resources using Binder-Oaxaca decomposition. Controls in 'Other explained' are in Table 1. (a) The standardized score on the Kessler-6 mental distress scale. (b) An indicator for Kessler-6 $\geq$ 13. (c) Standardized satisfaction with life overall (0–10 scale). (d) An indicator for agreement to the statement "I often feel very lonely". Full regression results are in Table D1 in S4 Appendix in S1 Online Appendix.

(0.91 sd) and are 14 ppts (85%) more likely to feel lonely. These gaps are similar if we reweight the samples to form mutually exclusive groups (see Fig D1 and Table D2 in S4 Appendix in S1 Online Appendix). We cannot causally attribute these gaps to deficits in psychosocial resources; however, decomposition analysis is informative in highlighting potential mechanisms and providing a sense of magnitude for the importance of gaps in psychosocial resources.

Collectively, people's psychosocial and cognitive resources account for 24–42% of the gaps in their wellbeing. This is either more than (mental distress, loneliness) or approximately the same as (high mental distress, life satisfaction) their demographics and family background combined. Locus of control is the single most important resource supporting mental wellbeing, which is consistent with the critical role of internal locus of control beliefs in emotional adjustment and the ability to handle stress generally, and at work in particular [47]. Emotional stability and sleep quality are key for mental distress, while social capital matters for loneliness. Other resources like cognition and risk tolerance have generally small independent effects.

## 4. Discussion

Previous efforts to measure certain psychosocial resources of extremely poor households have focused on developing countries [48,49]. In advanced economies, the sample of severely disadvantaged respondents in representative surveys is usually too small for them to be analyzed separately. Consequently, what we know about their resource deficits is generally based on

small, often non-representative, samples of people experiencing extreme poverty or homelessness. Researchers have found, for example, that homeless women living in shelters in Portland have social networks that include fewer people and involve less mutual giving and receiving (reciprocity) [50] (see also [51] for a review), while homeless individuals in Toronto receive less emotional and financial support from their social networks [52]. In Mexico City, extreme poverty has been linked to diminished internal locus of control and social support for coping [53]. Evidence of these resource deficits–albeit isolated–is important in light of the evidence that psychological and social resources may mediate the effect of poverty (or homelessness) on mental health, physical health, and emotional wellbeing [54,55].

Our research overcomes the data limitations of previous studies by analyzing disadvantaged individuals drawn from the universe of Australian administrative social assistance records, providing nationally representative evidence. A key contribution of our research is the extraordinary breadth of the psychological, social and cognitive resources that we consider. Previous studies have been more narrowly focused on these factors in isolation or in limited combinations.

Taken together, our results provide new evidence that vulnerable people living in wealthy countries have substantial deficits in multiple psychosocial resources–some of which are strongly associated with a reduction in their overall mental wellbeing. These deficits are apparent both between the vulnerable and the general population, and within the vulnerable population by degree of disadvantage.

Our focus on examining poverty through the lens of housing vulnerability is novel. In their recent review, [56] note that there has been a strong research emphasis on homelessness, arguing that the failure to take a broader perspective and consider all aspects of housing insecurity (e.g. housing affordability, stability, safety, etc.) has "resulted in much less being known about its true prevalence and the actual costs it imposes on society" (p. 95). Our results contribute to closing this gap by quantifying the large degree to which housing vulnerability is associated with increased mental distress and loneliness as well as a reduction in life satisfaction. We show that people's personal resources matter as much, or more, than their demographic characteristics and family background combined in driving the impact that housing vulnerability has on their diminished mental wellbeing.

It is also the case that, housing insecurity is related both to unhealthy behaviors and poor health outcomes [57]. Moreover, in wealthy countries, like Australia, where social and economic conditions are more favorable fewer people are likely to lack secure housing, while those who do may be particularly disadvantaged relative to the rest of society (see [58,59]). Policies targeting the health challenges stemming from housing vulnerability–or indeed disadvantage more generally–need to be cognizant of the constraints imposed by people's personality traits, risk preferences, control perceptions, cognition, and social capital.

In reaching these conclusions, we are cognizant of the limitations of our study. We recognize that the relationships between homelessness, resources and mental wellbeing are likely complex and bi-directional. While our study provides important descriptive evidence, given the observational nature of our data, our estimates cannot be interpreted as causal. We also acknowledge limitations in the comparison of resources across datasets owing to some differences in wording, scales and timing of questionnaires (see Table A1 in S1 Appendix in S1 Online Appendix), although there is no reason to think these differences should influence results in the direction of our conclusions.

Future research efforts should be directed towards testing whether psychosocial resources are malleable and have a causal role on the life outcomes of vulnerable people. If so, enhancing the psychosocial resources of vulnerable populations may be an effective way of improving their wellbeing.

## Supporting information

**S1 Online Appendix.**
(DOCX)

**S1 Replication files.**
(ZIP)

## Acknowledgments

We thank Tiffany Ho and Lihini De Silva for providing research assistance. This paper uses unit record data from the Household, Income and Labour Dynamics in Australia (HILDA) Survey. The HILDA Project was initiated and is funded by the Australian Government Department of Social Services (DSS) and is managed by the Melbourne Institute of Applied Economic and Social Research (Melbourne Institute). This paper also uses unit record data from the Journeys Home Survey, which is also funded by DSS and run by the Melbourne Institute. The findings and views reported in this paper, however, are those of the author and should not be attributed to either DSS or the Melbourne Institute.

## Author Contributions

**Conceptualization:** Deborah A. Cobb-Clark, Nathan Kettlewell.

**Data curation:** Nathan Kettlewell.

**Formal analysis:** Nathan Kettlewell.

**Funding acquisition:** Deborah A. Cobb-Clark.

**Investigation:** Deborah A. Cobb-Clark, Nathan Kettlewell.

**Software:** Nathan Kettlewell.

**Writing – original draft:** Deborah A. Cobb-Clark, Nathan Kettlewell.

**Writing – review & editing:** Deborah A. Cobb-Clark, Nathan Kettlewell.

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
