## [Decision Letter · Decision Letter 0]

27 Mar 2021

PONE-D-21-04348

Psychological, Social and Cognitive Resources and the Mental Wellbeing of the Poor

PLOS ONE

Dear Dr. Kettlewell,

Thank you for submitting your manuscript to PLOS ONE. After careful consideration, we feel that it has merit but does not fully meet PLOS ONE’s publication criteria as it currently stands. Therefore, we invite you to submit a revised version of the manuscript that addresses the points raised during the review process.

We look forward to receiving your revised manuscript.

Kind regards,

Abid Hussain

Academic Editor

PLOS ONE

Additional Editor Comments:

Dear Authors,

Please address the comments from reviewers.

Journal Requirements:

Reviewers' comments:

Reviewer's Responses to Questions

**Comments to the Author**

1. Is the manuscript technically sound, and do the data support the conclusions?

Reviewer #1: Partly

Reviewer #2: Yes

2. Has the statistical analysis been performed appropriately and rigorously? 

Reviewer #1: Yes

Reviewer #2: Yes

3. Have the authors made all data underlying the findings in their manuscript fully available?

Reviewer #1: No

Reviewer #2: No

4. Is the manuscript presented in an intelligible fashion and written in standard English?

Reviewer #1: Yes

Reviewer #2: Yes

5. Review Comments to the Author

Reviewer #1: The authors compare the 1138 individuals in an unusual Australian dataset of homeless people (the Journeys Home data) with the main representative longitudinal panel in Australia (the HILDA). They show how the homeless score worse in every domain imagineable and that only about 25% of the difference in mental wellbeing outcomes can be explained by their extensive battery of demographic and psychological measures. They show how worse degrees of homelessness are related to worse psychosocial characteristics.

The paper is competent and reads well, though is repetitive in places. At heart it is a descriptive paper that cannot make credible claims of causality and thus needs to make as interesting comparisons as possible. I have several recommendations that amount to axing what is of limited use and adding more that is of interest.

1. Axe Section 4.2 Its a re-run of Figure 3 that adds ittle and suffers from the problem that there are only 28 full homeless in the data.

2. Simplify Section 4.1 to show the average differences in important characteristics (pick your favourite 5 to 10) across the different degrees. Also calculate the implied distance between the 4 categories (ie how many times is the difference in a characteristic between secure 'non-trad' versus 'secure trad' the difference between 'non-secure' and 'secure traditional'). That gives the reader a sense of what the special group is and how different the groups really are.

3. Find a way to condense the four figures in Figure 3 to one graph/bar. At present it looks clunky and not memorable. You are trying to sell the Oaxaca decomposition here so make it memorable.

4. Change the labeling of the categories of variables. The word 'resources' is a very poor descriptor because it includes items that have for decades been called different things. Emotional stability and conscientiousness are two of the big-5 personality traits. Sleep is part of the GHQ12 notion of mental wellbeing itself (rather than a resource into it). I would just call them psychological characteristics and outcomes. Similarly, avoid causal language since many items put on the right-hand side could be on the left-hand side (so Figure 3 is a big stretch: why sleep is an input rather than an output is arbitrary).

5. Condense the literature sections. The reader mainly needs to know whether your descriptives are truly new: was it already known how much the homeless differ from the rest of the population in all those psychological dimensions?

I hence like the degree to which the paper informs me about the differences between the homeless and the rest, but I need to be told that this papers adds to what is known about that difference, and the paper needs to be shortened and sharpened to stick to those descriptives.

Reviewer #2: My general evaluation: the manuscript has been well prepared, with its methodological section being crafted sufficiently.

The study provides valuable insights that might contribute to policymaking towards homeless people in Australia. Therefore, I trust that the paper has merit to become a contribution to the literature. Nonetheless, I have some issues with the paper and believe they should be addressed before the paper can be ready for publication.

Please find below my specific comments, which might enable the authors to strengthen the paper and its flow of logic.

The findings are backed with sound methodology and sufficient sample size. And this is a positive aspect of the study. However, the research’s significance and values are not addressed adequately in the text.

The discussion lacks depth and needs further results’ clarification with existing policies regarding homelessness. The rising complexities of the problem have become far more difficult to measure than simplistic results we can see from a theoretical model or two.

I think some critical aspects should not be left out when we discuss the health issues of vulnerable people. Specifically: healthcare disparity (https://ajph.aphapublications.org/doi/abs/10.2105/AJPH.2013.301490, https://springerplus.springeropen.com/articles/10.1186/s40064-015-1279-x), health care system (https://ajph.aphapublications.org/doi/full/10.2105/AJPH.2005.076190), and healthcare discrimination (https://bmchealthservres.biomedcentral.com/articles/10.1186/1472-6963-14-376), to be specific.

As the authors stated, “The way that people perceive the world, process information, and make decisions is shaped by their psychological, social, and cognitive resources” I advise the authors to take a look at the Mindsponge mechanism (https://www.sciencedirect.com/science/article/abs/pii/S0147176715000826;
https://www.taylorfrancis.com/chapters/global-mindset-integration-emerging-socio-cultural-values-mindsponge-processes-quan-hoang-vuong/e/10.4324/9781315736396-8), which might provide theoretical support for the study’s objectives.

Figure 1’s stacked bar charts should be replaced by clustered bar charts for better clarity

The paper’s structure needs to be revised. Model 1 should be placed in the corresponding section where the main methodological discussions were provided.

Suppose the authors want to present mathematical models together with the results. In that case, I recommend the authors clearly address all the components of the model rather than keeping the general form (like Model 1).

Please specify the high-income country.

Last but not least, the study’s limitations are needed, please refer to this article for modern standards: https://www.nature.com/articles/d41586-020-01694-x

6. PLOS authors have the option to publish the peer review history of their article (what does this mean?). If published, this will include your full peer review and any attached files.

Reviewer #1: No

Reviewer #2: **Yes: **Quan-Hoang Vuong

---

## [Author Response · Author response to Decision Letter 0]

3 Jun 2021

We have included our response as an attachment to this submission.

---

## [Decision Letter · Decision Letter 1]

31 Jul 2021

PONE-D-21-04348R1

Psychological, Social and Cognitive Resources and the Mental Wellbeing of the Poor

PLOS ONE

Dear Dr. Kettlewell,

Thank you for submitting your manuscript to PLOS ONE. After careful consideration, we feel that it has merit but does not fully meet PLOS ONE’s publication criteria as it currently stands. Therefore, we invite you to submit a revised version of the manuscript that addresses the points raised during the review process.

The manuscript has significantly improved after revisions. There is till one outstanding minor issue on the use of term 'resources'. Authors are required to clarify this while responding to reviewer's comments.

We look forward to receiving your revised manuscript.

Kind regards,

Abid Hussain

Academic Editor

PLOS ONE

Journal Requirements:

Reviewers' comments:

Reviewer's Responses to Questions

**Comments to the Author**

1. If the authors have adequately addressed your comments raised in a previous round of review and you feel that this manuscript is now acceptable for publication, you may indicate that here to bypass the “Comments to the Author” section, enter your conflict of interest statement in the “Confidential to Editor” section, and submit your "Accept" recommendation.

Reviewer #1: (No Response)

2. Is the manuscript technically sound, and do the data support the conclusions?

Reviewer #1: Partly

3. Has the statistical analysis been performed appropriately and rigorously? 

Reviewer #1: Yes

4. Have the authors made all data underlying the findings in their manuscript fully available?

Reviewer #1: No

5. Is the manuscript presented in an intelligible fashion and written in standard English?

Reviewer #1: Yes

6. Review Comments to the Author

Reviewer #1: The authors have partly done what I asked and have improved the embedding in the literature, kicking out some of the superfluous analyses. I do find the text easier to read now. I appreciate they were not able to condense 4 graphs into one.

My main disagreement remains with the word 'resource'. I find that a misleading term as used in this paper, creating the false impression that differential 'resources' explains between 20-40% of the difference in mental wellbeing between the homeless and the general population. It is simply not true, and we should not want a false headline conclusion to be the take-home message of this paper.

The authors in their reply define 'resources' perfectly adequately. They claim resources are 'the qualities, skills, and support systems that assist people in achieving their goals and dealing with problems effectively" and that "our goal is to document the disparity in the resources that very vulnerable people have at their disposal". This all concords with the classic economic notion of a resource: something that can be drawn upon.

Sleep does not fit any of this. Sleep cannot be drawn upon, nor is it a skill, quality, or support system. It is an outcome that then further affects other parts of the person. So too can one argue that risk orientation is an outcome as extreme poverty changes the calculus on what can be achieved without risks.

If the authors want to retain the 'resources' label, which the authors seem hell-bent on, then they will have to exclude sleep and risk attitudes from that group when they make claims about how much is explained by resources. Of course, cognition and social capital too are affected by housing and the whole trajectory leading up to homelessness, but at least both can still be meaningfully interpreted as qualities and support that can be drawn upon, and hence a resource.

7. PLOS authors have the option to publish the peer review history of their article (what does this mean?). If published, this will include your full peer review and any attached files.

Reviewer #1: No

---

## [Author Response · Author response to Decision Letter 1]

12 Aug 2021

Please see response in cover letter attached with this submission.

---

## [Editor Report · Decision Letter 2]

28 Sep 2021

Psychological, Social and Cognitive Resources and the Mental Wellbeing of the Poor

PONE-D-21-04348R2

Dear Dr. Kettlewell,

We’re pleased to inform you that your manuscript has been judged scientifically suitable for publication and will be formally accepted for publication once it meets all outstanding technical requirements.

Kind regards,

Abid Hussain

Academic Editor

PLOS ONE
---

## [Editor Report · Acceptance letter]

1 Oct 2021

PONE-D-21-04348R2 

Psychological, Social and Cognitive Resources and the Mental Wellbeing of the Poor 

Dear Dr. Kettlewell:

I'm pleased to inform you that your manuscript has been deemed suitable for publication in PLOS ONE. Congratulations! Your manuscript is now with our production department. 

Kind regards, 

on behalf of

Dr. Abid Hussain 

Academic Editor

PLOS ONE